# How a Short-Lived Rumor of Residential Redevelopment Disturbs a Local Housing Market: Evidence from Hangzhou, China

Yanjiang Zhang [1,2], Hongyi Fan [2], Qingling Liu [2], Xiaofen Yu [1,2,*] and Shangming Yang [3]

1 Chinese Academy of Housing and Real Estate, Zhejiang University of Technology, No. 18 Chaowang Road, Gongshu District, Hangzhou 310014, China

2 School of Management, Zhejiang University of Technology, No. 18 Chaowang Road, Gongshu District, Hangzhou 310014, China

3 Hangzhou Jointfounder Information Technology Co., Ltd., Hangzhou 310012, China

* Correspondence: yxf@zjut.edu.cn

**Abstract:** This paper investigates how a short-lived rumor of residential redevelopment triggered herding trading and housing price overreactions in a local housing market in Hangzhou. Through event studies, we find that herding purchasing caused a short-term housing price overreaction. Simultaneously, existing homeowners became reluctant to sell, and the number of new listings for sale decreased temporarily. However, we find no evidence of a decrease in market efficiency. A herding investor who purchased an average home may have suffered a loss of CNY 593,907 after the rumor weakened, equivalent to 8.7 years of income for an average resident in Hangzhou in 2021. This study reveals the importance of government policy communication, and the detrimental impact of ambiguous urban renewal policies on housing market stability and wealth redistribution.

**Keywords:** residential redevelopment; housing price; herding; policy communication





## 1. Introduction

Housing price overreaction has generated severe economic and social problems, such as reducing housing affordability, increasing wealth inequality, and even causing economic recessions [1–3]. Housing price overreactions can be largely attributed to herding following an information cascade in an inefficient market [4–7]. As one form of information, rumors can cause price fluctuations, as they may lead investors' expectations and thus trading behaviors [8–10]. Especially in the Chinese institutional context (refer to Section 2.1 for more details), residential redevelopment projects and related rumors may disturb local housing markets [11–14]. However, there is little empirical evidence on how rumors of residential redevelopment contribute to housing price overreaction. In this paper, we tend to reveal how a short-lived rumor of residential redevelopment triggered herding trading and housing price overreactions.

This study takes a unique event in a local residential area of Hangzhou in China as a quasi-natural experiment (refer to Section 2.1 for details). In recent years, some decayed neighborhoods have been planned by the government to be renewed rather than redeveloped, but the renewed neighborhoods will lose the chance to be redeveloped by the government within at least 5 to 10 years. In January 2021, the People's Government of Xiacheng District of Hangzhou announced a neighborhood renewal program for the whole Zhaohui area, except for the residential project Zhaohui No. 6 (In January 2021, Hangzhou Xiacheng District People's Government issued a document "The 7th Plenary Session report of the 10th Session of the district Party committee", which mentioned the neighborhood renewal plans of each district in Zhaohui. According to the document, the scheme design of Zhaohui District 6 has not been completed due to the future community planning, causing a strong wave of belief of a redevelopment in the near future). The information was

interpreted by some housing investors as Zhaohui No. 6 was about to be demolished and rebuilt by the government. Importantly, the government has neither confirmed nor falsified the reconstruction expectation at the time of completion of our article (November 2022). For a long time, decayed neighborhoods have been attractive for investors to speculate about high redevelopment compensation from the government [15,16]. As shown in Figure 1, the local resale housing market was cold. After the neighborhood renewal plan was announced, a small group of investors, taking the renewal plan as a signal for residential redevelopment, bought the housing prices in Zhaohui No. 6 (the treatment project) up. The trading volume of the treatment project suddenly rose in January 2021, while the resale housing market in Hangzhou was apparently cold.

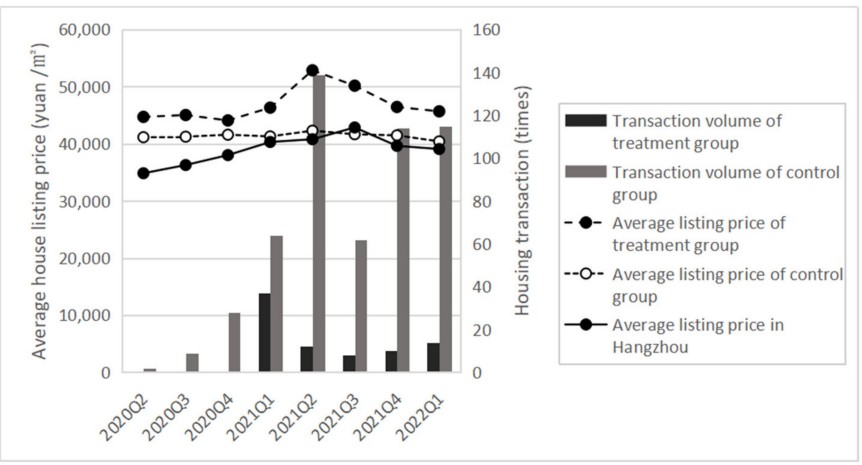

**Figure 1.** Real Estate Market in the Study Area from 2020Q2 to 2022Q1.

This study investigates how the group of housing investors herd over the rumor of residential redevelopment and disturb the local housing market in the Zhaohui area. Using the unique institutional setting in Hangzhou, China, we raise the following research questions. First, how did existing homeowners as sellers (hereafter, homeowners) respond to housing investors as buyers (hereafter, herding investors)? Second, did herding investors change market efficiency? Third, did herding investors profit from trading?

In this study, the short-lived rumor of residential redevelopment in Zhaohui No. 6 is regarded as an event that can be analyzed using the event study approach (refer to Section 3.2 for details). Our empirical analysis employs housing listings and transaction records in the resale housing markets of the Zhaohui area between June 2020 and March 2022. Through event studies, we reach the following findings. The local market was just temporarily boomed up by the herding investors. Specifically, the demand shock formed by the herding investors buying housing properties prompted the market to reach a new equilibrium after a three-month adjustment period. In particular, rumor and herding investors caused an overreaction in housing prices, triggered the supply response of the local housing market, and temporarily curbed the owners' willingness to sell. Finally, our results suggest that the information shock did not change market efficiency since the volatility of listing prices did not change. Furthermore, by the end of the studied period, we estimate that the herding buyers lost as much as RMB 593,907, equivalent to the income of an average resident of Hangzhou for 8.7 years (The income data is from the Hangzhou Bureau of Statistics of China: http://tjj.hangzhou.gov.cn/art/2021/3/18/art_12292796 82_3852554.html (accessed on 16 February 2023). According to the Hangzhou Bureau of Statistics, the average disposable income of an urban resident of Hangzhou in 2020 is 68,666 yuan) (refer to Section 4 for details).

Our study makes the following contributions. First, this paper contributes to the literature on urban redevelopment [11,17,18]. The existing studies on urban redevelopment focus mainly on externalities related to neighborhood integration influenced by redevelopment projects [18–20]. This study investigates urban redevelopment from a new angle, i.e.,

it provides empirical evidence on whether and how an unconfirmed residential redevelopment project affects neighborhood housing prices. Second, our empirical findings add to the understanding of how herding investors cause housing price overreaction [21–23]. Most housing herding literature studies herding in terms of housing buying [5,6]. This study distinguishes itself by showing how existing homeowners as sellers respond to housing investors as buyers. Second, we offer some important insights into how rumors can influence housing prices. Evidence from several empirical studies has established that rumors can affect asset prices [9,10,24], while studies on how rumors cause real estate price fluctuations are rare.

Furthermore, it adds to the literature that studies the unintended effects of government policies [25–28]. We uncover a mechanism by which ordinary neighborhood renewal plans generate expectations of residential demolition, leading to fluctuations in housing prices. Thus, this paper also relates to the literature on expectations affecting housing prices [22,29,30].

The remainder of the paper is organized as follows: Section 2 introduces the institutional and literature backgrounds. Section 3 details the data and empirical design. Section 4 presents and discusses the results. Section 5 concludes the paper.

## 2. Background

### 2.1. Institutional Background

Aged homes in decayed neighborhoods in China's large cities are attractive for investors. Because they tend to be located in the central areas of China's large cities, they are often demolished and redeveloped by the government [31–34], and homeowners can usually receive monetary compensation of over 50% higher than the market prices for giving up property rights and resettlement [11,29–37].

This study focuses on nine aged residential projects with names ranging from Zhaohui No. 1 to Zhaohui No. 9 that were built in the 1880s. They are located in the central area of Hangzhou, which is one of the 16 largest cities in China, with a population over 10 million (The population data is from the Seventh National Population Census of China. Population of a city is defined as residents who register their residence in the city (differentiating from the part of people who temporarily visit the city)). The building structures and the internal neighborhood facilities of the nine projects are homogenously designed, constructed, and maintained. The building blocks are all brick and concrete structures, and they are less than six floors in height and without a lift. The location distribution of these nine projects is shown in Figure 2. Except for Zhaohui No. 5, Zhaohui No. 6, and Zhaohui No. 8, which are close to a university, the location features of the nine projects are basically the same.

The recent neighborhood renewal policies in Hangzhou have led some optimistic investors to believe that Zhaohui No. 6 will be redeveloped by the government (Rumors about the residential redevelopment of Zhaohui No. 6 were sourced from relevant media reports. Please refer to the link https://baijiahao.baidu.com/s?id=1695350392063444870&wfr=spider&for=pc, https://baijiahao.baidu.com/s?id=1695065980838504296&wfr=spider&for=pc, and https://baijiahao.baidu.com/s?id=1695916824368197075&wfr=spider&for=pc for details (accessed on 16 February 2023)). In January 2021, the local government released neighborhood renewal plans for the neighborhoods in Zhaohui Street. All the remaining eight residential projects (Zhaohui No. 1–5, Zhaohui No. 7–9) have renewal plans except Zhaohui No. 6. Some home investors have interpreted this information as a sign that Zhaohui No. 6 will soon be demolished and redeveloped by the government because the projects that have been planned or have the possibility of redevelopment will not be renewed. The redevelopment plan of residential projects is always kept confidential. By the time this paper is submitted (November 2022), the redevelopment expectation is neither confirmed nor falsified by the government.

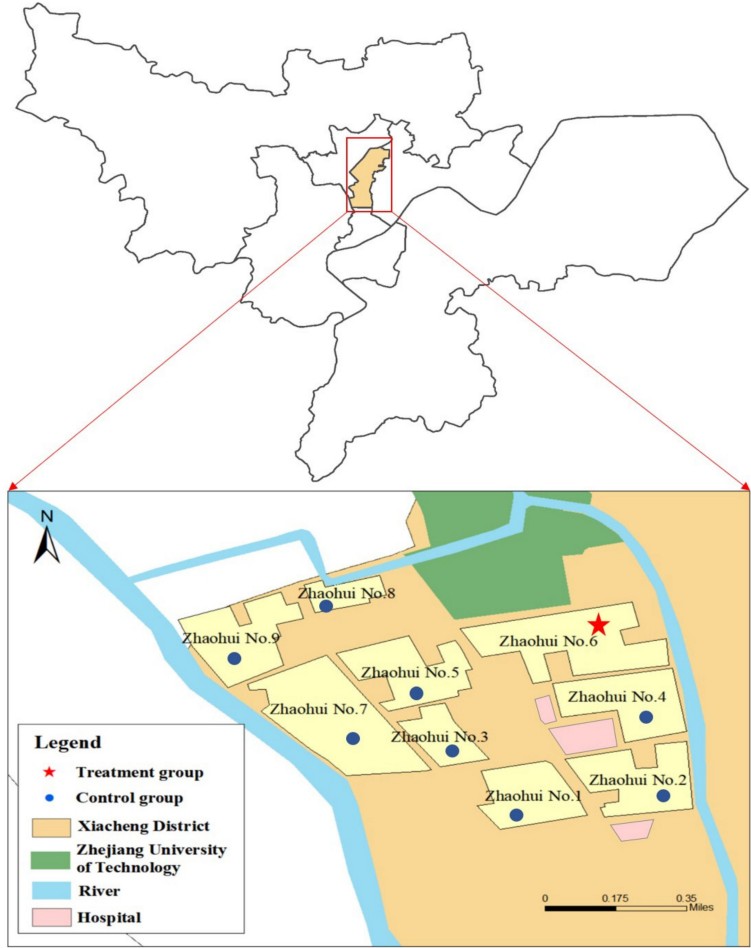

**Figure 2.** Location of the nine studied residential projects in Zhaohui Street of Hangzhou, China.

As shown in Figure 1, compared with the other eight Zhaohui projects (the control group), the housing listing prices of Zhaohui No. 6 (the treatment group) surged two months after the unconfirmed expectation was stimulated (the left vertical line, Jan 2021). We observe an overreaction of listing prices of Zhaohui No. 6, and then the listing prices returned close to the average level after the weakening of the residential redevelopment rumor.

### 2.2. Literature Background

2.2.1. Previous Studies on Residential Redevelopment

Residential redevelopment aims to improve the housing conditions and living environment of a city [12,18,38]. Housing externalities have been extensively studied in the literature on residential redevelopment [39,40]. Studies have shown that residential redevelopment increases the property values of renewed properties as well as properties in neighboring areas [17,40–42]. Meanwhile, externalities of urban redevelopment also make property owners reluctant to pay to renew their own properties, as there is no incentive to spend money so that others can benefit [39,41,43–45].

To date, few studies have investigated the impact of a rumor of residential redevelopment on housing prices. Baek and Jin (2021) used the difference-in-differences method to examine the impact of urban regeneration projects on neighborhood housing prices in Taipei, and they reported positive effects just after the announcement of the plan. In a recent study on residential redevelopment projects [46], Liang et al. (2020) showed that redevelopment projects may generate an anticipatory psychological effect among consumers and investors, which is often reflected in housing prices in advance [29].

However, all of these studies investigate the impact of a confirmed residential redevelopment or redevelopment plan on housing prices. The influence of unconfirmed or a rumor of redevelopment on property prices is insufficiently studied, especially under the institutional background of China.

2.2.2. Previous Studies on Price Overreaction and Rumors

A large and growing body of literature has investigated how heterogeneous investors contribute to the price overreactions of a whole housing market [3,21,23,47–50]. Bayer et al. (2021) argued for a behavioral contagion mechanism by which uninformed investors enter the real estate market following other investors' investment activities, causing prices to overreact [23]. Using data from the housing market in Singapore, Fu and Qian (2014) demonstrated that short-term speculators lead to an overreaction in housing prices through momentum trading [21]. Zhou (2016) argued that long-term investors contribute to market efficiency because they overreact to policies less than consumers [3].

As a form of information, rumors can influence price dynamics. Currently, empirical research on whether and how rumors contribute to housing price fluctuations is scarce. A notable exception is Kiel and McClain (1995), who found that rumors about negative facility siting trigger housing price responses [51]. Most literature examines how rumors' rise, spread, and fade are capitalized into prices and cause stock price changes [9,10,24,52]. For example, a study by Ahern and Sosyura (2015) found that rumors can lead to stock price overreactions and subsequent reversals due to investors' inability to judge the veracity of media information [9]. Andrei and Cujean (2017) showed that rumors can generate stock price increases or reversals through word-of-mouth communication [24]. Schmidt (2020) revealed that unconfirmed rumors can affect stock prices, and that short investment horizons can facilitate information sharing, thereby accelerating the capitalization of information into stock prices [10].

2.2.3. A Short Summary

From all the studies reviewed, it can be found that, first, far too little attention has been given to the influence of unconfirmed residential redevelopment projects on housing prices. Second, whether and how rumors contribute to housing price overreactions has not been thoroughly studied. While previous research has shown that rumors can cause herding trading, and thus asset price fluctuations, empirical evidence on how rumors can cause housing price overreactions remains limited. Using the unique institutional setting in Hangzhou, China, this research thus tends to fill these gaps by revealing how a short-lived rumor of residential redevelopment triggered herding trading and housing price overreactions.

## 3. Methodology

This section may be divided by subheadings. Section 3.1 succinctly describes the data used in the study and the setting of variables. Section 3.2 details the empirical design adopted in this study and the experimental conclusions that can be drawn.

*3.1. Data Source and Variable Definitions*

This study used housing listing and resale transaction data from a major real estate agency (Lianjia) in China (Lianjia is a major real estate agency in China. Please refer to the link https://lianjia.com/ (accessed on 16 February 2023) for details). Our data cover a full set of resale housing listing records from June 2020 to March 2022 for 9 aged residential projects, namely, ranging from Zhaohui No. 1 to Zhaohui No. 9. Each listing or transaction record contained the housing unit listing/transaction price, date of listing/transaction, and hedonic features such as floor level, size, and layout. Table 1 shows variable definitions, and Table 2 presents simple summary statistics.

**Table 1.** Variable definitions.

| Variable | Definition |
|---|---|
| Log(Price) | The logarithm of listing price per square meter of a housing unit; |
| Trans_Ratio | The ratio of number of transactions in each month to total units of each neighborhood, in ‰; |
| List_Ratio | The ratio of number of new listings in each month to total units of each neighborhood, in ‰; |
| Volatility | Volatility of listing prices for each neighborhood in each month; |
| Post | A binary variable that equals 1 if data observation is after the information shock, otherwise it equals 0; |
| Treat | A binary variable that equals 1 if a housing unit or neighborhood belongs to the treatment group (Zhaohui No. 6), 0 if a housing unit or neighborhood belongs to the control group (the other 8 Zhaohui projects); |
| Log(Size) | The logarithm of the size of the listed housing unit; |
| Bedroom | Number of bedrooms; |
| Living_room | Number of living rooms; |
| Furnished | A dummy variable that equals 1 if the observed housing unit is well furnished, otherwise it equals 0; |
| **Floor** | A vector of dummy variables indicating whether the observed housing unit is at the lower, middle or high floor level; |

Note: It is a unique feature that the building structures and internal neighborhood facilities of the 9 projects are homogeneously designed, constructed, and maintained. Thus, we consider size, layout, furnished, and floor range over other hedonic features. Selection of control variables can be found in Clapp et al. (2012), Fu and Qian (2014), and Bao et al. (2020) [15,21,35].

**Table 2.** Descriptive statistics of the variables.

| Panel A: Individual-Level Dataset | | | | | |
|---|---|---|---|---|---|
| **Variable** | **Obs.** | **Mean** | **Std.** | **Min.** | **Max.** |
| Treat = 1 | | | | | |
| Log(Price) | 139 | 10.73 | 0.1 | 10.52 | 11.02 |
| Post | 139 | 0.44 | 0.5 | 0 | 1 |
| Log(Size) | 139 | 4.03 | 0.16 | 3.72 | 4.3 |
| Bedroom | 139 | 2.13 | 0.38 | 1 | 3 |
| Living_room | 139 | 1.02 | 0.35 | 0 | 2 |
| Furnished | 139 | 0.35 | 0.48 | 0 | 1 |
| Floor | 139 | 1.14 | 0.78 | 0 | 2 |
| Treat = 0 | | | | | |
| Log(Price) | 859 | 10.62 | 0.08 | 10.37 | 11.26 |
| Post | 859 | 0.58 | 0.49 | 0 | 1 |
| Log(Size) | 859 | 4.04 | 0.15 | 3.71 | 4.31 |
| Bedroom | 859 | 2.1 | 0.46 | 1 | 3 |
| Living_room | 859 | 1.06 | 0.34 | 0 | 2 |
| Furnished | 859 | 0.33 | 0.47 | 0 | 1 |
| **Floor** | 859 | 1.13 | 0.77 | 0 | 2 |
| Panel B: Neighborhood-Level Dataset | | | | | |
| **Variable** | **Obs.** | **Mean** | **Std.** | **Min.** | **Max.** |
| Treat = 1 | | | | | |
| Trans_Ratio | 22 | 1.02 | 1.75 | 0 | 8.04 |
| List_Ratio | 22 | 2.05 | 1.53 | 0.28 | 6.93 |
| Post | 22 | 0.68 | 0.48 | 0 | 1 |
| Volatility | 21 | 0.23 | 0.06 | 0.07 | 0.33 |
| Treat = 0 | | | | | |
| Trans_Ratio | 176 | 1.21 | 1.33 | 0 | 6.43 |
| List_Ratio | 176 | 2.44 | 1.52 | 0 | 7.55 |
| Post | 176 | 0.68 | 0.47 | 0 | 1 |
| Volatility | 176 | 0.18 | 0.13 | 0.01 | 0.91 |

*3.2. Empirical Design*

Event studies mainly examine the abnormal changes in target variables after a specific event occurs, which is widely used in market-based empirical research in finance and accounting [53–56]. The basic logic of conventional event study is to first define the event window, then calculate the abnormal returns in the event window (the yield gap between observed return and expected return in the absence of the event) and the cumulative abnormal returns, and finally measure the significance of the event impact using statistical tests of both indicators.

This paper is concerned with how an optimistic group of herding investors, as buyers, respond to the short-lived rumor of residential redevelopment, and whether their purchases can disrupt the local housing market after the information shock. As discussed in Section 2, the neighborhood renewal plan for Zhaohui Street was interpreted by some housing investors as Zhaohui No. 6 was about to be demolished and redeveloped by the government. In this study, the short-lived rumor of residential redevelopment in Zhaohui No. 6 was regarded as an event that can be analyzed using the event study approach. Given that the study period coincides with the COVID-19 pandemic, COVID-19 has had some impact on the real estate market. Our approach did not fully apply the event study methods used in finance. We used the regression model instead of the direct mean difference to measure the abnormality of data in the sample, housing listing prices in particular. Once the coefficient of the key explanatory variable is statistically significant after the event, it can be concluded that the event has real treatment effects.

From January 2021, the neighborhood renewal plan for the neighborhoods in Zhaohui Street started circulating online, sparking discussion among the crowd. Considering the lag of the official release of the policy, we took the month before the government issued the neighborhood renewal plan (i.e., December 2020) as the time of the information shock, the event date. Furthermore, we selected 15 months after the event occurrence date as the event window period (January 2021 to March 2022). We chose six months before the event window as the forecast period (June 2020 to November 2020). The empirical design consists of four steps.

In the first part, we used event study methodology to assess when and how herding investors as buyers respond to the residential redevelopment rumor triggered by the neighborhood renewal policy. Specifically, we tested the change in transaction ratios after the information shock. We conducted the event study as shown in Equation (1).

$$Trans\_Ratio_{it} = \beta + \sum_{k=1}^{T_1} \alpha_k Pre_k \times Treat_i + \sum_{k=1}^{T_2} \gamma_k Post_k \times Treat_i + \omega_t + \mu_i + \varepsilon_{it} \quad (1)$$

The explained variable is the ratio of the number of new transactions to the total units of neighborhood $i$ in month $t$; $Treat_i$ is a dummy variable that equals 1 if neighborhood $i$ belongs to the treatment group and 0 otherwise. $\omega_t$ and $\mu_i$ denote the time fixed effect by year-month and neighborhood fixed effect, respectively. $\varepsilon_{it}$ is the residual term.

The coefficients $\alpha_k$ and $\gamma_k$ capture the trend differences between the treatment groups and the control groups in each event month relative to the benchmark period (December 2020). We included 6 months prior to the benchmark period ($\sum_{k=1}^{T_1} \alpha_k Pre_k \times Treat_i$) and all available 15 months thereafter ($\sum_{k=1}^{T_2} \gamma_k Post_k \times Treat_i$).

In the second part, an event study was developed to investigate homeowners' reactions to herding investors. We applied housing listing prices rather than contract prices in the analyses, as the listing price reflects the willingness to sell by existing homeowners (the sellers). To formally analyze the impact of the herding purchase behavior of housing investors on housing prices, we set up a regression equation, which is specified as Equation (2).

$$Log(Price)_{jit} = \beta + \sum_{k=1}^{T_1} \alpha_k Pre_k \times Treat_i + \sum_{k=1}^{T_2} \gamma_k Post_k \times Treat_i + \delta X_{it} + \omega_t + \mu_i + \varepsilon_{it} \quad (2)$$

where $Log(Price)_{jit}$ denotes the logarithm of the unit price of each listed housing unit $j$ in neighborhood $i$ in each year-month $t$. We controlled for a vector of variables $X_{it}$, consisting of Log(Size), Bedroom, Living Room, Floor and Furnished, and $\varepsilon_{it}$ is the residual. The definitions of the rest of the control variables and the coefficients are the same as those in Equation (1).

In the third part, we conducted the event study to capture the impact of herding investors on willingness to list for sale of homeowners, formalizing a regression model as follows:

$$List\_Ratio_{it} = \beta + \sum_{k=1}^{T_1} \alpha_k Pre_k \times Treat_i + \sum_{k=1}^{T_2} \gamma_k Post_k \times Treat_i + \omega_t + \mu_i + \varepsilon_{it} \quad (3)$$

The explained variable is the ratio of the number of new listings to the total units of neighborhood $i$ in month $t$. The definitions of the rest of the control variables and the coefficients are the same as those in Equation (1).

In the last part, we investigated whether the room for bargaining changed after the information shock. We predicted that the difference between transaction prices and listing prices in treated groups dramatically increased after the herding purchasing of optimistic housing investors, while the bargaining room would remain stable in control groups. Specifically, we used the T test to observe the abnormal changes in the bargaining room during the window period.

## 4. Empirical Results and Discussion

### 4.1. Results

Column (1) in Table 3 reports the estimation applying Equation (1), which takes *Trans_Ratio* as the dependent variable. As shown in column (1), we test for the impacts of the residential redevelopment rumor triggered by the neighborhood renewal policy on housing transactions. The coefficients of $Pre_k \times Treat$ are statistically nonsignificant at the conventional levels, which indicates that there was no significant difference in housing transactions between treated groups and control groups in the pre-event period. The building structures and the internal neighborhood facilities of the 9 projects are homogenously designed, constructed, and maintained. Thus, Zhaohui No. 6 and all the remaining 8 residential projects (Zhaohui No. 1–5, Zhaohui No. 7–9) were reasonably similar in terms of housing liquidity before the government released the neighborhood renewal plan of Zhaohui Street. In contrast, the coefficients of $Post_1 \times$ Treat and $Post_3 \times$ Treat are 2.18 and 6.14, respectively; the former is significant at the 10% level, while the latter is significant at the 1% level. The results suggest that the housing transaction ratio in treated groups increased evidently in the first and third months after the information shock, which supports the finding that herding investors as buyers were attracted by the redevelopment expectation of Zhaohui No. 6 after the announcement of the neighborhood renewal plan. They took the renewal plan as a signal of residential redevelopment, immediately digested existing listed housing units in Zhaohui No. 6, and suddenly increased the housing transaction ratio.

**Table 3.** Impacts on transaction and listing behaviors.

| Time Window | (1) | (2) | (3) |
|---|---|---|---|
| | Trans_Ratio | Log(Price) | List_Ratio |
| (June 2020) Pre6 × Treat | 0.4069 | 0.0523 | −0.1235 |
| | (0.310) | (1.445) | (0.071) |
| (July 2020) Pre5 × Treat | 0.3080 | 0.0658 * | −0.1125 |
| | (0.235) | (1.923) | (0.065) |
| (August 2020) Pre4 × Treat | 0.3500 | 0.0727 * | −1.1186 |
| | (0.267) | (1.956) | (0.642) |
| (September 2020) Pre3 × Treat | 0.4500 | 0.0406 | 1.4536 |
| | (0.343) | (1.236) | (0.834) |
| (October 2020) Pre2 × Treat | −0.0015 | 0.0337 | 2.5622 |
| | (0.001) | (1.112) | (1.470) |
| (November 2020) Pre1 × Treat | 0.1710 | 0.0450 | −0.9466 |
| | (0.130) | (1.243) | (0.543) |
| (January 2021) Post1 × Treat | 2.1775 * | 0.0264 | −0.8259 |
| | (1.661) | (0.710) | (0.474) |
| (February 2021) Post2 × Treat | 0.2816 | 0.0890 | −1.8984 |
| | (0.215) | (1.208) | (1.089) |
| (March 2021) Post3 × Treat | 6.1369 *** | 0.1944 *** | −2.9245 * |
| | (4.681) | (5.080) | (1.678) |
| (April 2021) Post4 × Treat | −1.8179 | 0.2613 *** | 0.2804 |
| | (1.387) | (7.160) | (0.161) |
| (May 2021) Post5 × Treat | −0.5724 | 0.1408 ** | −2.3331 |
| | (0.437) | (2.532) | (1.338) |
| (June 2021) Post6 × Treat | 0.5359 | 0.2485 *** | −1.6222 |
| | (0.409) | (5.206) | (0.930) |
| (July 2021) Post7 × Treat | −0.1393 | 0.1311 *** | −0.6469 |
| | (0.106) | (3.162) | (0.371) |
| (August 2021) Post8 × Treat | 0.0491 | 0.1965 *** | 0.0668 |
| | (0.037) | (5.187) | (0.038) |
| (September 2021) Post9 × Treat | 0.6724 | 0.1800 *** | −1.3150 |
| | (0.513) | (3.224) | (0.754) |
| (October 2021) Post10 × Treat | −0.1155 | 0.1042 ** | 0.2651 |
| | (0.088) | (2.300) | (0.152) |
| (November 2021) Post11 × Treat | −1.5269 | 0.1236 ** | 0.0108 |
| | (1.165) | (2.179) | (0.006) |
| (December 2021) Post12 × Treat | 0.3884 | 0.0876 * | 0.1439 |
| | (0.296) | (1.936) | (0.083) |
| (January 2022) Post13 × Treat | 0.4557 | 0.0886 | −0.2566 |
| | (0.348) | (1.489) | (0.147) |
| (February 2022) Post14 × Treat | −0.5063 | 0.1287 *** | −1.4335 |
| | (0.386) | (2.670) | (0.822) |
| (March 2022) Post15 × Treat | 0.0141 | 0.1022 ** | −0.5642 |
| | (0.011) | (2.471) | (0.324) |
| Furnished | | 0.0332 *** | |
| | | (7.033) | |
| Log(size) | | −0.0650 *** | |
| | | (3.318) | |
| Bedroom | | 0.0039 | |
| | | (0.653) | |
| Living_Room | | 0.0178 ** | |
| | | (2.520) | |
| Cons. | 1.1474 *** | 10.8598 *** | 2.4586 *** |
| | (9.701) | (154.822) | (15.634) |
| **Floor** | / | YES | / |
| Project FE | YES | YES | YES |
| Year_Month FE | YES | YES | YES |
| N | 198 | 998 | 198 |
| R-squared | 0.601 | 0.426 | 0.415 |

Note: Robust standard errors are clustered at the neighborhood level and are reported in parentheses; *** $p < 0.01$, ** $p < 0.05$, * $p < 0.1$.

Column (2) in Table 3 presents the estimated regression results of Equation (1) using the natural log of the listing price as the dependent variable. The coefficients of $Pre_k \times Treat$ are almost statistically insignificant. However, the coefficients of the interaction of Post with Treat are positive and statistically significant at least at the 10% level, except for the

coefficient of $Post_{13} \times$ Treat, which is not significant. We find that the interaction coefficient between Post and Treat first increases and then decreases over time, reaching the highest in the fourth post-event period. Specifically, the coefficient of $Post_4 \times$ Treat is 0.26 and statistically significant at the 1% level. There are three major implications for our event study results. First, there were no significant differences in housing listing price dynamics between Zhaohui No. 6 and the other 8 residential projects in the pre-event period. Second, after the information shock, the listing price of the housing units in the treated groups increased substantially, which indicates that the rumors of residential redevelopment drove the relative increase in the housing listing price of Zhaohui No. 6. It also means that the owners of Zhaohui No. 6 believed in the expectation of residential redevelopment, and raised their listing prices one after another. Third, potential housing unit sellers listed their housing units after learning and digesting the information, and the unconfirmed expectation of residential redevelopment of Zhaohui No. 6 was capitalized in market prices and maximized in the fourth month after the rise of the residential redevelopment rumor.

Column (3) shows the regression results of Equation (3) for testing the impact of herding investors on willingness to list for sale of homeowners, which takes *List_Ratio* as the dependent variable. The coefficient of $Post_3 \times$ Treat is −2.92 and significant at the 10% level. The result implies that the listed housing units in Zhaohui No. 6 experienced a volume decline relative to the other eight residential projects in the third month after the rise of the residential redevelopment rumor. Together with the results in column (2), it indicates that, after seeing the crazy trading behavior of herding investors, a group of potential housing unit sellers of Zhaohui No. 6 raised housing prices, while a group of existing homeowners of Zhaohui No. 6 bet on the redevelopment, and stopped selling properties in the short term.

In Table 4, we test for abnormal changes in the bargaining room during the window period. The estimated result of the event window from 1 April 2021 to 31 May 2021, is significant at the 1% level, suggesting that there was a significant difference between the bargaining room of the real estate market in Zhaohui No. 6 and other residential projects in Zhaohui Street four months after the rise of the residential redevelopment rumor. Clearly, the expected change in homeowners was driven by herding investors, leading to a significant increase in the price difference between transaction prices and listing prices until a few months after the information shock. In Table 4, we also notice that the trading volume of Zhaohui No. 6 temporarily surged after the rise of the residential redevelopment rumor, and fell back after absorbing the investment demand shock.

**Table 4.** Impacts on bargaining room.

| Event Window | Average Price Difference of Zhaohui No. 6 (Number of Transactions) | Average Price Difference of Other Residential Projects in Zhaohui Street (Number of Transactions) | Mean Diff. | *p* Value |
|---|---|---|---|---|
| [1 January 2021–31 January 2021] | 1.085 (8) | 1.074 (12) | 0.012 | 0.748 |
| [1 March 2021–31 March 2021] | 1.087 (23) | 1.056 (17) | 0.031 | 0.238 |
| [1 April 2021–31 May 2021] | 1.173 (6) | 1.050 (23) | 0.124 *** | 0.0012 |

Note: Robust standard errors are clustered at the neighborhood level and are reported in parentheses; *** $p < 0.01$.

Collectively, our results show how the group of housing investors herding over and residential redevelopment rumors impact the local housing market over time. In January 2021, the government announced a neighborhood renewal program for the whole Zhaohui area, except for the residential project Zhaohui No. 6, which triggered a short-lived rumor that Zhaohui No. 6 was about to be demolished and rebuilt by the government. A small group of herding investors took the lead in responding, quickly digesting the existing listed housing units of Zhaohui No. 6, and temporarily increasing the ratio of the number of

transactions. Then, the existing homeowners in Zhaohui No. 6 temporarily postponed housing listings and raised listing prices in response to the herding investors. In the end, as the rumor was never confirmed, the residential redevelopment expectation gradually faded, and the market returned to normal.

*4.2. Discussion*

4.2.1. Market Efficiency

Did herding investors change market efficiency? Thus far, we have revealed the responses of herding investors to residential redevelopment rumors in terms of their housing transacting and listing behaviors. To examine the subsequent impacts on market efficiency, we adopted a reflection index of market efficiency, exploring the volatility of the listing prices. The settings of the event day, estimation window, and event window are the same as those in Equation (1). Specifically, we performed the following regression:

$$Volatility_{it} = \beta + \sum_{k=1}^{T_1} \alpha_k Pre_k \times Treat_i + \sum_{k=1}^{T_2} \gamma_k Post_k \times Treat_i + \omega_t + \mu_i + \varepsilon_{it} \quad (4)$$

The explained variable is the volatility of listing prices of neighborhood $i$ in month $t$; $Treat_i$ is a dummy variable that equals 1 if neighborhood $i$ belongs to the treatment group and 0 otherwise. $\omega_t$ and $\mu_i$ denote the time-fixed effect by year-month and neighborhood fixed effect, respectively. $\varepsilon_{it}$ is the residual term. We also included 6 months prior to the benchmark period ($\sum_{k=1}^{T_1} \alpha_k Pre_k \times Treat_i$) and all available 15 months thereafter ($\sum_{k=1}^{T_2} \gamma_k Post_k \times Treat_i$).

Figure 3 plots the coefficients of the interaction terms between $Treat_i$ and each event month dummy. As shown in Figure 3, the estimated coefficients are generally insignificant for months during the whole event period, which means that herding investors did not significantly change the market efficiency. One interpretation of this finding is that local market information is largely transparent.

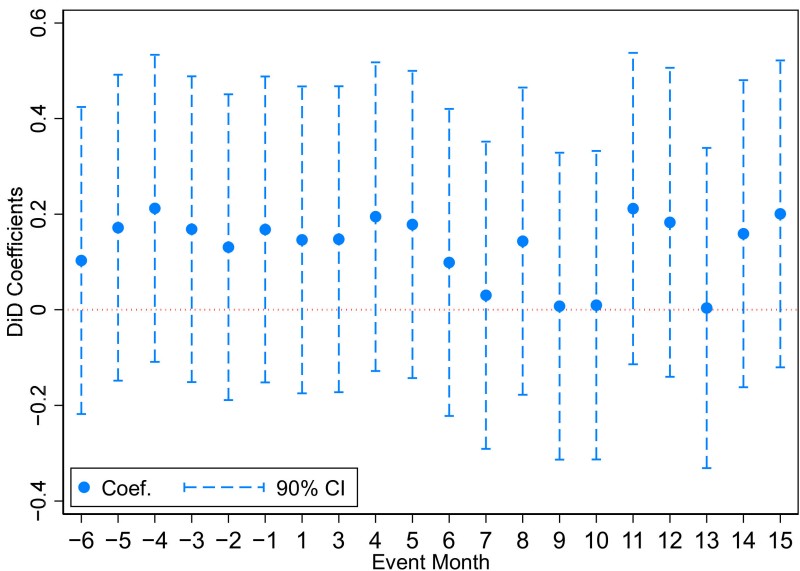

**Figure 3.** The impact of herding investors on price volatility.

4.2.2. Wealth Effect

From the above analysis, it can be seen that the rise of the residential redevelopment rumor and herding investors as a demand shock led to the overreaction of housing listing prices of Zhaohui No. 6. That is, housing prices fell back after a brief surge driven by herding investors. Next, we investigate who made money out of the residential redevelop-

ment rumor. Assuming that each housing unit in Zhaohui No. 6 is 60 square meters, the reconstruction expectation will remain the same (neither confirmed nor falsified). If the value of housing purchased by each investor is calculated based on the last event period (March 2022), the profit situation is shown in Table 5.

**Table 5.** Profit and loss of herding investors over time.

| The Housing Purchase Time | Gains (Chinese Yuan) |
| --- | --- |
| (January 2021) | 112,778 |
| (February 2021) | −249,007 |
| (March 2021) | −396,027 |
| (April 2021) | −593,907 |
| (May 2021) | −290,407 |
| (June 2021) | −436,795 |
| (July 2021) | −295,999 |
| (August 2021) | −384,795 |
| (September 2021) | −313,177 |
| (October 2021) | −115,111 |

Note: The estimation applies the results from Tables 2 and 3.

As summarized in Table 5, for optimistic herding investors who bought housing units in Zhaohui No. 6 for the first time after the information shock (January 2021), the investment gained approximately RMB 100,000 after the rumor weakened. Conversely, for optimistic herding investors who bought housing units in Zhaohui No. 6 within 2 to 10 months after the information shock (from February 2021 to October 2021), the losses on this investment ranged from RMB 200,000 to RMB 600,000. At its most extreme, we find that a herding investor who purchased an ordinary home in Zhaohui No. 6 in the fourth month after the information shock (April 2021) lost RMB 593,907 at the end of the studied period, equivalent to 8.7 years of income of an average resident in Hangzhou in 2021.

## 5. Conclusions

This study investigates how a short-lived rumor of government-initiated residential redevelopment impacts the local housing market over time. Specifically, the rumor that Zhaohui No. 6 was about to be demolished and redeveloped by the government provides a unique natural experiment. Using an event study specification for housing units in the one treated group (Zhaohui No. 6) and eight control groups (Zhaohui No. 1–5, Zhaohui No. 7–9), we reveal the responses of herding investors and homeowners to the residential redevelopment rumor in terms of their housing transacting and listing behaviors, as well as the subsequent impacts on market efficiency.

Our empirical results show that herding investors quickly bought out the existing listed housing units in Zhaohui No. 6 following the redevelopment rumor. The demand shock formed by herding investors pushed the market adjustment to a new equilibrium after a three-month adjustment period. We also document that, following the speculative housing purchases of such herding investors, the listing price of housing units in Zhaohui No. 6 showed an overreaction, even though the local resale housing market was cold. Our findings further suggest that the rumor following the announcement of the neighborhood renewal plan and herding investors collectively led to changes in housing transacting and listing behaviors, which not only increased the listing price in Zhaohui No. 6, but also reduced the owners' willingness to sell. In contrast, we find that herding investors did not affect market efficiency since the volatility of listing prices did not change. Moreover, we quantify the return of herding investors in the housing investment of Zhaohui No. 6, and most herding investors suffered huge losses after the rumor weakened.

Overall, the findings of this research trigger some critical policy implications. First, we find that ambiguous urban renewal policies can lead to public confusion and misunderstanding, as well as the short-term herding of home purchases by investors, resulting in abnormal fluctuations in the housing market. However, the Chinese government usually

has no incentive to respond to rumors. The fundamental reason is that the cost of dispelling rumors is not commensurate with the benefit. Therefore, the results of the study should make policymakers pay attention to the importance of government policy communication, and force the government to create a direct communication channel with the public to listen to their concerns. Second, the results suggest that the redistribution of wealth via urban renewal may exacerbate inequalities, and advocates should become more aware of the economic risks inherent in urban renewal. Furthermore, we note that residential redevelopment rumors and herding investors can undermine the stability of the local housing market. The Chinese government's lack of response is likely to leave many people uncertain about the future of their housing properties. Thus, the government needs to maintain the continuity and stability of real estate policies while constantly maintaining the stability of market expectations.

**Author Contributions:** Conceptualization, Y.Z. and X.Y.; methodology, Y.Z. and S.Y.; software, H.F.; validation, X.Y., Y.Z., Q.L. and S.Y.; formal analysis, H.F.; investigation, Y.Z.; resources, X.Y.; data curation, Y.Z. and H.F.; writing—original draft preparation, H.F. and X.Y.; writing—review and editing, Q.L. and S.Y.; visualization, Y.Z.; supervision, X.Y.; project administration, X.Y.; funding acquisition, X.Y. All authors have read and agreed to the published version of the manuscript.

**Funding:** This research was funded by the National Natural Science Foundation of China, awarded to Yanjiang Zhang, Grant No. 72004203; National Natural Science Foundation of China, awarded to Xiaofen Yu, Grant No. 72274176; Ministry of Education Key Projects of Philosophy and Social Sciences Research (CN), awarded to Xiaofen Yu, Grant No. 18JZD033; Fundamental Research Funds for the Provincial Universities of Zhejiang, Grant No. GB201901002.

**Data Availability Statement:** This study used housing listing and resale transaction data from a major real estate agency (Lianjia) in China.

**Conflicts of Interest:** The authors declare no conflict of interest.

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
