# Peer review of "How a Short-Lived Rumor of Residential Redevelopment Disturbs a Local Housing Market: Evidence from Hangzhou, China"

_land, doi:10.3390/land12020518_

Round 1
Reviewer 1 Report
A very interesting study investigating how a rumor can influence the local housing market within a short time period with a uncommon perspective. I appreciate the author’s creativity. However, this paper lacks the fundamental background introduction and has to significantly improve English editing. I suggest the paper should be published after major revision.
General comments:
1. The introduction section looks a little bit odd as it directly goes to story-telling while does not provide enough knowledge background to the audience.
2. It lacks the fundamental introduction of literature investigating a similar topic, i.e., how rumors can influence the housing market. A detailed literature review should be added.
3. Contradictory statements are found. For example, in line 122, the authors said the rumor was not either confirmed or falsified. In line 127, the authors said the rumor ended.
4. Informal expressions in section 3.2.
5. The author should briefly introduce the logic/theory behind their overarching methodology design.
6. There are only 6 indicators in the hedonic price model. Is that enough? Why did the authors choose these indicators?
7. Isn’t section 4.2.1 a part of the results?
8. It is interesting to explore why the Chinese government does not respond to the rumors. What kind of possible reasons may lead to this?
Detailed comments:
1. English should be improved throughout the manuscript. Some long sentences are very hard to read. The use of tenses is messy.
2. Reference format is inconsistent. Please check.
3. The quality of Figure 1 can be improved. Now it is very dizzy.
4. Line 77, no reference.
5. Figure 2 lacks a compass and scale.
6. Not enough evidence to support section 2.
7. Please provide a link for the Lianjia agency as a reference.
8. What does Return in Table 5 mean?

Reviewer 2 Report
Whilst this paper contains original research of relevance to the journal, it is poorly situated in the literature. Previous literature is mentioned briefly and how this might add to this literature. This however is not adequately demonstrated. Indeed, the paper barely extends beyond the case. Conceptually the paper is weak. This paper would required a complete re-write and as such I recommend the paper is rejected. The case may have some merit but the author(s) have not adequately demonstrated how it builds on an adds to the international literature.
Reviewer 3 Report
I have reviewed the manuscript entitled “How a Short-Lived Rumour of Residential Redevelopment Disturbs a Local Housing Market: Evidence from Hangzhou, China”.
Dear authors, the paper presents an interesting study with a strong scientific impact. The results of this study are beneficial for research, industry professionals, and policymakers.
The empirical method used seems reasonable and appropriate to illustrate the phenomenon. The data and results are comprehensively represented. The conclusion section is well structured, summarizing and clearly interpreting the data.
However, I have some comments:
- To begin with, the correct term in the title is "rumor," not "rumour".
- The abstract does not summarize the article well; the problem to be addressed and the techniques are not defined.
- The introduction presents in detail the structure of the article, gives the reader a clear idea of what to expect in the following sections but does not describe the methodology of the survey.
- Regarding the article's citations and bibliography, the sources used are numerous but a wider range of bibliographic sources on residential redevelopment should be included.
- The empirical analysis employs housing listings and transaction records in the resale housing markets of the Zhaohui area between June 2020 and March 2022. The period considered coincides with the COVID19 pandemic. However, I would recommend that the authors emphasize the fact that COVID19, which the authors did not mention, had a significant impact on property values as well as on the material uncertainty of valuation and consequently on the real estate valuation methodology used.
- Improve the illustration of the data sample considered provide more information to the reader about Zhaohui Area No. 6.
- Page 6, Table 2, please do not separate one table into two different pages.
Please complete the paper in this regard.
Round 2
Reviewer 1 Report
This study quantitatively explores how rumours can influence housing prices whilst using a relatively small case with several buildings. The transferability to other scenarios is in question. However, that’s the nature of this study, so I will not argue.
The authors addressed my concerns very well for the rest of my comments.
